# How useful is a complete urinary tract ultrasound in orchiepididymitis?

Eva Aeschimann[ID][1]*, Oliver Sanchez[ID][2], Jacques Birraux[3], Barbara E. Wildhaber[3], Sergio Manzano[1]

1 Pediatric Emergency Division, Department of Pediatrics, Gynecology, and Obstetrics, Geneva University Hospitals, University of Geneva, Geneva, Switzerland, 2 Division of Child's and Adolescent's Surgery, Department of Pediatrics, Gynecology, and Obstetrics, University Center of Pediatric Surgery of Western Switzerland, Lausanne, Switzerland, 3 Division of Child's and Adolescent's Surgery, Department of Pediatrics, Gynecology, and Obstetrics, University Center of Pediatric Surgery of Western Switzerland, University of Geneva, Geneva, Switzerland

* eva.aeschimann@hcuge.ch

**Data Availability Statement:** The dataset is now available in the Yareta public repository (DOI https://doi.org/10/gnqmw9).

**Funding:** The authors received no specific funding for this work.

## Abstract

Orchiepididymitis (OE) is a frequent cause of pediatric emergency department attendance in boys presenting with acute scrotum. The etiology of most episodes of OE remains unclear and there is no consensus regarding the correlation between OE and underlying genitourinary malformations. Whether imaging evaluation should comprise complete urinary tract ultrasonography (US) or voiding cystography is a subject of debate. The aim of this retrospective, single-center study was to analyze i) the number/type of urinary tract malformations detected by US following a first episode of OE in boys with no previously known malformation and ii) the frequency of associated urinary tract infection (UTI). We reviewed the records of 495 boys <16 years presenting to our pediatric emergency department with acute scrotum between January 2012 and December 2017. Patients with incomplete radiological data were excluded. Of 119 boys with a radiologically-confirmed first episode of OE, 99 had a complete urinary tract US and were included in the study. No genitourinary malformation was detected (0%). Urinary cultures showed UTI in 3/98 (3.1%) patients. Mean age at presentation was 9.7 years (standard deviation, 3.9) with a three-peak incidence of OE at 10–13 years, 4–5 years, and during infancy. *Conclusion*: Complete urinary tract US does not appear to be useful during a first episode of OE in countries with an antenatal US screening rate similar to Switzerland. The very low UTI rate suggests that a urinalysis is sufficient to investigate a first episode of OE and antibiotics should be reserved for positive urinalysis only.

## Introduction

Orchiepididymitis (OE) is an inflammation of the testicles and epididymis. In boys less than 16 years old, the etiology differs from the adult form and remains poorly understood. Current literature offers many possible pathophysiological explanations such as a post-infectious inflammatory condition [1,2], viral infection [2,3], bacterial infection from the urinary tract

**Competing interests:** The authors have declared that no competing interests exist.

**Abbreviations:** OE, orchiepididymitis; CRP, C-reactive protein; SD, standard deviation; US, ultrasonography; UTI, urinary tract infection.

[3–5], pre-existing urinary tract malformation [3–12] or vesical dysfunction [3,4,10,12], trauma, auto-immune disease or vasculitis [12], or idiopathic origin [4,12,13]. The link with an underlying urinary tract malformation, such as vesicoureteral reflux, ectopic ureters, prostatic utricle, posterior urethral valves, meatal stenosis, urethral stenosis, urinary reflux in the genital tract, is still a subject of debate. Some studies suggest a correlation between these urinary tract anomalies and OE [3–9,12], whereas others refute this hypothesis [1,14–16]. Thus, it is difficult to recommend specific investigation guidelines, such as urinary analysis, testicular and/or urinary tract and kidney ultrasonagraphy (US), or micturating cysto-urethrogram. This debate also impacts on guidelines for antibiotic treatment, which is either systematically recommended [14,17], or varies with the age of the patient [18,19], or with the association of a urinary tract infection (UTI) [2,4,19].

In our pediatric emergency department, local practice guidelines currently mandate to investigate every boy presenting a suspected OE with i) a blood test, ii) an urinalysis and iii) a testicular, urinary tract and kidney US. All children diagnosed with OE receive a prescription of paracetamol. Regarding antibiotic treatment, it is limited to urinary tract infections (positive urinalysis).

The primary objective of our study was to analyze the benefit of urinary tract and renal US in the initial work-up of a patient presenting with a first episode of OE, by evaluating the rate of new urinary tract malformations found by means of this imaging. Our hypothesis was that the proportion of patients with an unknown urinary tract malformation detected during a first episode of OE would be low and therefore that these imaging exams are of little benefit. This study could therefore contribute to avoid unnecessary exams and achieve greater medical efficiency. The secondary objectives were i) to determine the proportion of UTI associated with OE and ii) to evaluate inflammatory markers of patients presenting with OE and iii) to compare these to patients excluded due to a known urinary tract malformation. Finally, we aimed to describe clinical symptoms and signs of OE and their frequency.

## Methods

### Study design and participants

We conducted a retrospective cohort study of boys aged less than 16 years of age who consulted the pediatric emergency department of a tertiary hospital with a diagnosis of OE between 1 January 2012 and 31 December 2017. Patients were identified in the electronic medical records using the following keywords: orchiepididimytis, orchitis, acute scrotum, testicular pain, testicular torsion, testicular trauma, hydatid of Morgagni, inguinoscrotal hernia, spermatic cord cyst, in the "reason of consultation" or "diagnosis" fields. The study started in January 2012 as it is the year that medical records started to be digitalized at our institution.

All patients with a confirmed first episode of OE diagnosed by US and not meeting exclusion criteria, were included in the primary analysis. Exclusion criteria were: a known urinary tract malformation, past history of OE, and those who did not have a complete record, i.e. urinary tract and kidney, US work-up during a consultation for a suspicion of OE. The flowchart for study inclusion is shown in Fig 1. Note that each selected OE episode corresponds to a unique patient as opposed to excluded episodes of OE, e.g. due to a known urinary tract malformation.

### Excluded patients subgroup analyses

Patients with OE and a previously diagnosed urinary tract malformation were combined to form a first subgroup, and those with diagnosed OE who did not have a urinary tract and kidney US in the work-up as a second subgroup.

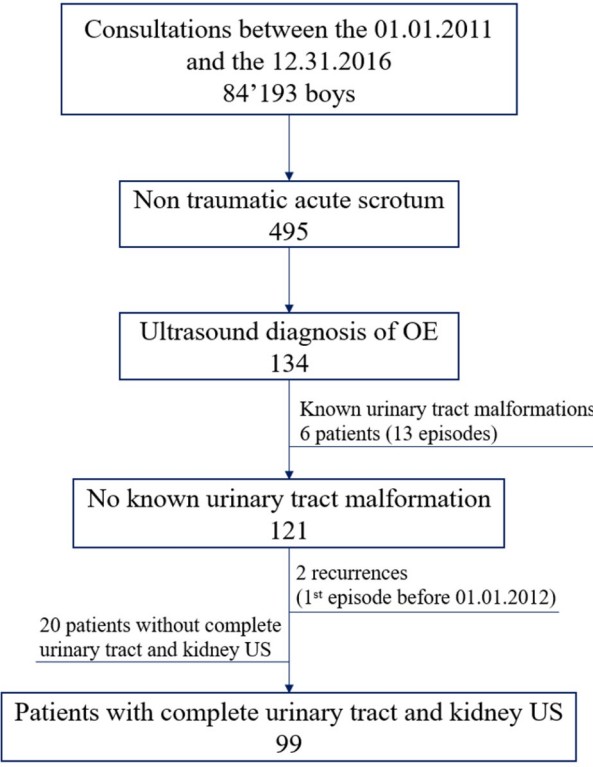

**Fig 1. Patient selection flowchart.** Patient inclusion and exclusion criteria applied in the study with number of included patients below each criterion.

## Protocol

Patient records were reviewed retrospectively for numerous parameters including data for each case description, medical history, clinical presentation, blood tests (white blood cell count, CRP), urinary analysis (urinalysis and culture) and US examination (testicular, urinary tract and kidney). The following definitions were used: 1) OE: inflammation of the epididymis and/or the testicles diagnosed either by i) an increase in blood flow and lack of signs for testicular torsion by a Doppler US or ii) by surgical exploration. 2) Urinary tract malformations: vesicoureteral reflux, ectopic ureter, prostatic utricle, posterior urethral valves, urethral stenosis, meatal stenosis, urethral duplication. 3) UTI: presence of leucocytes (more than traces) or positive nitrite in urinalysis and a urine culture with the presence of pathogenic bacteria $\geq$ 50000 CFU/ml [20]. 4) Acute scrotum: unilateral or bilateral testicular pain with or without scrotal swelling, redness or local heat.

## Outcomes

The primary outcome was the number of newly discovered urogenital malformations diagnosed by renal and urinary tract US after a first OE episode. Secondary outcomes were the number of UTI associated with an OE episode; the medical history and clinical description of patients presenting with an OE episode; the incidence peak/s of OE in our population; and the blood and urinary work-up results related to the presence or absence of UTI and their diagnostic performance.

## Statistical analysis

Normally distributed data were expressed using mean and standard deviation (SD). Fisher's test was used to compare categorical data. Student's t-test was used to compare continuous variables. A confidence interval of 95% was calculated using the Clopper-Pearson test. A *p*-value <0.05 was considered as significant. Statistical analysis were performed using the SPSS V.25 program (SPSS Inc, Chicago, USA).

## Ethics statement

Approval was obtained from the Geneva Scientific Research and Ethics Commission (CCER2017-00030). The Scientific Research and Ethics Commission delivered the authorization to the principal investigator to access medical records without informed consent in accordance with the Swiss Human Research Act (article 34) [21] to exclusively obtain specific information related to OE. All medical information was collected in a confidential and anonymized database solely accessible by the principal investigator and her research team. The procedures used in this study adhere to the tenets of the Declaration of Helsinki.

## Results

Between January 2012 and December 2017, 84,193 boys less than16 years old consulted our pediatric emergency department. Of these, 495 presented with non-traumatic acute scrotum. A diagnosis of OE was made in 134 cases (27.1%). Six (4.7%) patients were excluded as they had a known diagnosed urinary tract malformation. Ninety-nine patients presenting with a first OE episode and a complete urinary tract and renal US examination were included in the study (Fig 1). Mean age was 9.7 years (SD 3.9). The youngest patient was aged 2 months and the oldest 16 years. Three incidence peaks by age appeared: <1 year, 4–5 years and 10–13 years (Fig 2). Concerning the primary outcome, no urinary tract malformation was diagnosed by US: 0/99 (0.0%; 95% CI, 0–3.7).

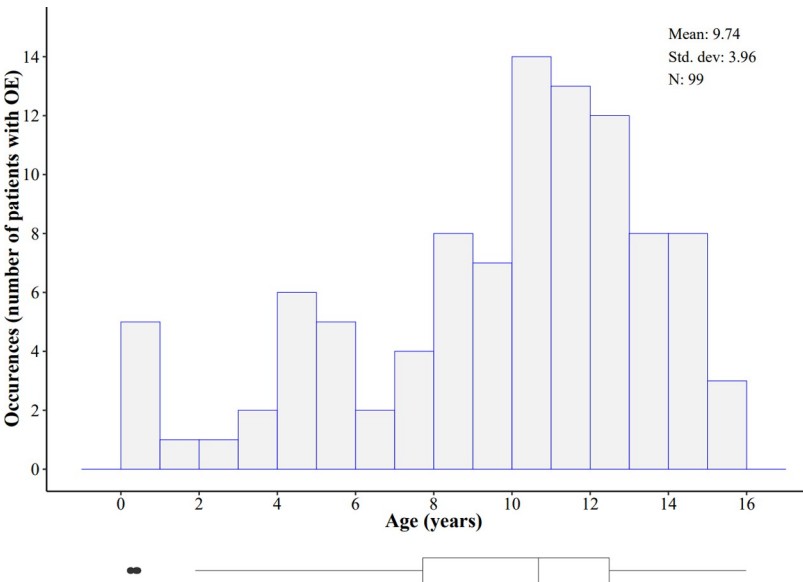

**Fig 2. Age distribution of patients.** Histogram and boxplot representing the age distribution of the patients with orchiepididymitis included in the study.

### Medical history

No fever was found in 94/99 patients (94.9%; 95% CI, 88.6–98.3). One child with a history of fever had a UTI. No patient presented with urethral discharge and only 14/86 patients had associated abdominal pain (16.3%; 95% CI, 9.2–25.8). Dysuria was present in 7/99 cases (7.1%; 95% CI, 2.9–14.0), but all urinary tests were negative for infection.

### Clinical examination

Testicular pain was present in 97/99 cases (98.0%; 95% CI, 92.9–99.8) and laterality was consistently correlated with radiological finding. Scrotal redness was present in 60/90 patients (66.7%; 95% CI, 55.9–76.3) and scrotal swelling in 64/98 cases (65.3%; 95% CI, 55.0–74.6). Testicular induration was only found in one (1/97) patient (1.0%; 95% CI, 0–5.6) and cremasteric reflex was absent in 22/80 cases (27.5%; 95% CI, 18.1–38.6).

### Follow-up

Seventy-seven of 93 patients had a follow-up after 48 h (82.8%; 95% CI, 73.6–89.8). Among these, 19/77 presented with persistent symptoms (25.0%; 95% CI, 15.8–36.3).

### Laboratory and imaging studies

A urinalysis was performed in 98/99 patients; none was positive for nitrites. Five of 98 patients presented with leucocytes (5.1%; 95% CI, 1.7–11.5) and 3/98 had a documented UTI (3.1%; 95% CI, 0.6–8.7). Urinalysis performance to detect UTI is shown in Table 1. A complete blood count was performed in 88 cases. The mean white blood cell count was 9.0 G/L (SD, 0.4). CRP was performed in 83 children with a mean of 6.4 ng/ml (SD, 2.0). Sensitivity, specificity, positive and negative predictive values and likelihood ratio of these tests for diagnosing UTI are described in Table 1.

The three children presenting an OE with a UTI were between 3 and 4 months old. One child had a history of fever. Concerning investigation work-up, these children had leucocytes on urinalysis and urinary culture was positive for an *Escherichia coli* infection (Table 2).

Six patients were excluded because of a previously known urinary tract malformation and accounted for 13 consultations (4.7%) (Table 3). Presence of leucocytes on urinalysis (>traces) and diagnosed UTI were statistically more frequent in the subgroup of patients with a pre-existing diagnosis of urinary tract malformation than in included patients (Tables 3 and 4).

**Table 1. Relationships between laboratory results and the presence of urinary tract infection.**

| | Sensitivity | Specificity | PPV [1] | NPV [2] | LR+ [3] | LR- [4] |
|---|---|---|---|---|---|---|
| **Leukocytes on urinalysis $\geq$ 1+** | 100% (95% CI 33.9–100.0) | 97.9% (95% CI 95.8–97.9) | 60,0% (95% CI 20,4–60,0) | 100% (95% CI 97.9–100.0) | 47.5 (95% CI 8.1–47.5) | 0.0 (95% CI 0.0–0.7) |
| **Leukocytosis $\geq$ 11 g/L***  | 100% (95% CI 31.8–100.0) | 81.2% (95% CI 78.8–81.2) | 15,8% (95% CI 5.0–15.8) | 100% (95% CI 97.0–100.0) | 5.3 (95% CI 1.5–5.3) | 0.0 (95% CI 0.0–0.9) |
| **CRP $\geq$ 19 mg/L*** | 66.7% (95% CI 13.0–98.2) | 93.8% (95% CI 91.7–94.9) | 28.6% (95% CI 5.6–42.1) | 98.7% (95% CI 96.6–99.9) | 10.7 (95% CI 1.6–19.4) | 0.36 (95% CI 0.02–0.95) |

[1] positive predictive value;

[2] negative predictive value

[3] positive likelihood ratio

[4] negative likelihood ratio.

* best cut-off value.

**Table 2. Included patients with urinary tract infection.**

|           | Age       | CRP (mg/L) | Blood leukocytes (absolute no.) | Leukocytes on urinalysis | Urine culture        |
|-----------|-----------|-----------|----------------------------------|--------------------------|----------------------|
| Patient 1 | 3 months  | 0         | 12.5                             | 3 +                      | *E. coli* $>10^5$    |
| Patient 2 | 4 months  | 19        | 11.0                             | 3 +                      | *E. coli* $>10^5$    |
| Patient 3 | 4 months  | 20        | 11.6                             | 3 +                      | *E. coli* $10^5$     |

## Discussion

Our findings show that no urinary tract malformation was diagnosed in any of the boys presenting with a first episode of OE during the 6-year study period. We thus conclude that there is no benefit in performing a urinary tract and kidney US for every patient presenting with a first OE episode in countries with generalized prenatal US screening. This conclusion is in line with the desire to rationalize unnecessary examinations and treatments in medicine. Our findings are similar to the results of a recent article by Lee et al. [22]. However, the usefulness of urinary tract and kidney US for the detection of a urinary tract malformation associated with OE remains a subject of debate in the published literature. Some authors recommend a urinary tract and renal US in every case as they consider OE to be strongly associated with genitourinary tract malformation [6,7,23], while others recommend this type of investigation only in the case of relapse [4,8,14,16], associated UTI [8,22] or according to the age group [7,22]. For the latter reason, some recommend a urinary tract and renal US only in patients less than one year of age [7,22]. Lee et al. [22] also suggested some risk factors, such as age less than one year-old and positive urinary culture, that should lead to investigation for a urinary tract malformation.

During our study period, we excluded 6 (4.7%) children (accounting for 13 OE episodes) due to a previously known urogenital malformation. If we had included these episodes, the malformation rate would have been 11.6%. This is a smaller proportion compared with what is found in the medical literature. Other studies found a urogenital malformation of 18% to 47%, depending on the population included [6,7,12,16,22,24]. What we share with those who looked for it, is that most if not all urogenital malformation where known before the OE episode [12,22].

There is an important variability in the literature regarding the associated UTI rate. By using the American Academy of Pediatrics definition for UTI [20], our study showed that the frequency of UTI associated with OE was very low (3.1%) and this result is in line with several studies [4,13,14,25]. Haecker et al. [14] reported a positive urinary culture in 2/49 (4.1%) patients, Santillanes et al. [25] in 4/97 (4.1%) cases and Graumann et al. [4] did not find any positive culture in 63 children. However, Merlini et al. [7] and Siegel et al. [6] showed a strong association between UTI and OE (40% in both studies). This difference may be due to the population studied. Children included in these studies had a high rate of urogenital malformation, mainly in the less than one-year-old population. Therefore, they showed a strong relation

**Table 3. Comparison between included patients and patients excluded with a known urogenital malformation.**

|                                           | Included patients mean (no.) | Excluded patients mean (no.) | *p*-value  |
|-------------------------------------------|------------------------------|------------------------------|------------|
| **Temperature (°C)**                      | 36.6 (98)                    | 36.5 (12)                    | 0.07       |
| **Leukocytosis (absolute no.)**           | 9.0 (88)                     | 8.1 (11)                     | 0.37       |
| **CRP (mg/L)**                            | 6.4 (83)                     | 10.5 (11)                    | 0.47       |
| **Urinalysis $\geq$ 1+ leukocyte: no. (%)** | 5/98 (5.1%)                  | 9/13 (69.2%)                 | **<0.001** |
| **Urinary infection: no. (%)**            | 3/98 (3.1%)                  | 4/13 (30.8%)                 | **0.003**  |

**Table 4. Details of the 6 patients with a previously known urogenital malformation.**

| | No of incidences | Age | Urinary infection | Urogenital malformation |
|---|---|---|---|---|
| **Patient 1** | 1 | 7 years | Ø | Right vesicoureteral reflux<br>Posterior urethral valves |
| **Patient 2** | 3 | 9 months<br>11 months<br>17 months | Ø<br>Yes (*Pseudomonas*)<br>Ø | Anorectal malformation:<br>Imperforate anus with proximal rectourethral fistula |
| **Patient 3** | 6 | 4 years<br>5 years<br>9 years<br>9.2 years<br>9.3 years<br>9.7 years | Ø<br>Ø<br>Yes (*E. coli*)<br>Ø<br>Yes (*E. coli*)<br>Yes (*E. coli*) | Complex uropathy:<br>Right single kidney with double system<br>Dysfunctional bladder with urethral duplication<br>Kidney failure |
| **Patient 4** | 1 | 15 years | Ø | History of testicular rhabdomyosarcoma |
| **Patient 5** | 1 | 7 years | Ø | Severe hypospadias |
| **Patient 6** | 1 | 17 months | Ø | Right kidney double system with bilateral vesicoureteral reflux |

between UTI and urogenital malformation with a high positive predictive value of malformation if a UTI was diagnosed.

A comparison of the results obtained in the first group of included patients and the subgroup of excluded patients with a previously known urogenital malformation showed a significant difference in the UTI rate, which was 3.1% in the former group and 30.8% in the second group. An existing urogenital malformation is therefore a risk factor for presenting with a concomitant UTI during an OE episode. Our results are comparable with other studies [16,22,24], which recommend to investigate children with a UTI by urinary tract and kidney US to search for a possible urogenital malformation for this reason. In patients without a previously documented malformation and a first episode of OE, urinary analysis and urinary culture should be performed for work-up, but an antibiotic treatment should not be introduced unless a UTI is suspected on the urinary analysis [19,25]. Given the low prevalence of UTI in these patients, blood test markers (white blood cell count and CRP) are not considered useful for work-up given their poor diagnostic performance.

One of the secondary objectives was to evaluate the clinical presentation of OE. We observed that children presented most of the time with pain. On the other hand, redness and scrotal swelling were less frequent than expected in patients with OE as they were reported in only two-thirds of cases. Moreover, the presence of cremasteric reflex does not appear to be a good diagnostic criterion as it was absent in 27.5% of our cases. Interestingly, as this is not well described in the literature, 75% of our patients described symptom regression in the first 48 h following the first medical consultation at the pediatric emergency department.

Our study has some limitations. First, the retrospective study design renders it more susceptible to selection bias. Second, results are based on the review of medical records, which can be incomplete. However, most cases followed local recommendations. Due to our study design, our results can only be generalized to countries such as Switzerland where antenatal follow-up is widespread and thus most urogenital malformations are diagnosed before birth. We acknowledge that US is only one of many available diagnostic imaging exams for renal and urinary tract malformations and cannot exclude all malformations. However, as our primary objective of our study was to evaluate the utility of this specific imaging technique, we do not consider that this limitation compromises the results. Our secondary objective was the evaluation of the patient group known for urogenital malformation, which was small. Even if our results are statistically significant, these cannot be generalized as two of the six children presented the majority of OE episodes.

To conclude, our study shows that a complete urinary tract US does not appear to be useful in children less than16 years old presenting with a first episode of OE in countries with a widespread antenatal US screening program. This is reinforced by the fact that only 4.7% of all OE had a previously known urogenital malformation. In addition, as the rate of UTI was very low, we consider that only a urinalysis is sufficient to investigate a first episode of OE and antibiotics should be reserved for positive urinalysis only.

## Author Contributions

**Conceptualization:** Eva Aeschimann, Oliver Sanchez, Jacques Birraux, Barbara E. Wildhaber, Sergio Manzano.

**Data curation:** Eva Aeschimann.

**Formal analysis:** Eva Aeschimann, Oliver Sanchez, Sergio Manzano.

**Investigation:** Eva Aeschimann.

**Methodology:** Eva Aeschimann, Barbara E. Wildhaber, Sergio Manzano.

**Project administration:** Eva Aeschimann.

**Supervision:** Oliver Sanchez, Jacques Birraux, Barbara E. Wildhaber, Sergio Manzano.

**Validation:** Oliver Sanchez, Barbara E. Wildhaber, Sergio Manzano.

**Visualization:** Sergio Manzano.

**Writing – original draft:** Eva Aeschimann.

**Writing – review & editing:** Oliver Sanchez, Jacques Birraux, Barbara E. Wildhaber, Sergio Manzano.

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
