## [Decision Letter · Decision Letter 0]

8 Nov 2021

PONE-D-21-31965How useful is a complete urinary tract ultrasound in orchiepididymitis?PLOS ONE

Dear Dr. Aeschimann,

Thank you for submitting your manuscript to PLOS ONE. After careful consideration, we feel that it has merit but does not fully meet PLOS ONE’s publication criteria as it currently stands. Therefore, we invite you to submit a revised version of the manuscript that addresses the points raised during the review process.

In trying to understand this improvement, the role of treatment received from the ER in terms of antibiotic and pain management is unclear. Was there a consistent ER protocol for treatment? Were antibiotics given only when the urinalysis was positive? Was there a standard pain management regimen? The 3 documented UTIs occurred in infants less than 6 months of age. Although they had negative urinary tract ultrasounds, would you advise further evaluation with a VCUG in these patients after the first OE episode?

We look forward to receiving your revised manuscript.

Kind regards,

Israel Franco, M.D.

Academic Editor

PLOS ONE

Journal Requirements:

 [No funding was received to assist with the preparation of this manuscript.]

[All authors certify that they have no affiliations with or involvement in any organization or entity with any financial interest or non-financial interest in the subject matter or materials discussed in this manuscript.]

Additional Editor Comments:

please address reviewers questions for acceptance

In trying to understand this improvement, the role of treatment received from the ER in terms of antibiotic and pain management is unclear. Was there a consistent ER protocol for treatment? Were antibiotics given only when the urinalysis was positive? Was there a standard pain management regimen? The 3 documented UTIs occurred in infants less than 6 months of age. Although they had negative urinary tract ultrasounds, would you advise further evaluation with a VCUG in these patients after the first OE episode?

Reviewers' comments:

Reviewer's Responses to Questions

**Comments to the Author**

1. Is the manuscript technically sound, and do the data support the conclusions?

Reviewer #1: Yes

Reviewer #2: Yes

2. Has the statistical analysis been performed appropriately and rigorously? 

Reviewer #1: Yes

Reviewer #2: I Don't Know

3. Have the authors made all data underlying the findings in their manuscript fully available?

Reviewer #1: Yes

Reviewer #2: Yes

4. Is the manuscript presented in an intelligible fashion and written in standard English?

Reviewer #1: Yes

Reviewer #2: Yes

5. Review Comments to the Author

Reviewer #1: The submitted article presents useful data for clinical practice. The submitted statistical analysis is valid, showing diagnosis of GU tract anomalies to be rare. I do not routinely obtain upper tract imaging in patients presenting with orchitis, but there is debate about this, especially in younger boys. The submitted article adds to the evidence that upper tract imaging is not needed.

Reviewer #2: In their retrospective review of 495 boys less than 16 years old presenting to the ER with signs and symptoms of non traumatic acute scrotum over a six year period from 2012 - 2017, the authors reported on 99 boys with their first episode of OE diagnosed by US who had no previously known urinary tract malformation and who also had a complete urinary tract US to rule out an underlying urinary tract malformation. The primary aim was to determine the yield of the screening “complete urinary tract US” in this patient population. None of these ultrasounds identified any underlying urinary tract malformation.

I found this to be a well written and clinically useful manuscript. In addition to the screening urinary tract US data, other findings in this patient population included a very low incidence of UTI (3.1%), inflammatory markers that were indeterminate and noncontributory, and 75% of patients had improved symptoms after 48 hours. In trying to understand this improvement, the role of treatment received from the ER in terms of antibiotic and pain management is unclear. Was there a consistent ER protocol for treatment? Were antibiotics given only when the urinalysis was positive? Was there a standard pain management regimen?

The 3 documented UTIs occurred in infants less than 6 months of age. Although they had negative urinary tract ultrasounds, would you advise further evaluation with a VCUG in these patients after the first OE episode?

Minor corrections: The study dates given in the Abstract are 2011-2017 while in the Methods and Results the dates are 2012-2017. Also, in figure 1 the dates are 2011-2017.

6. PLOS authors have the option to publish the peer review history of their article (what does this mean?). If published, this will include your full peer review and any attached files.

Reviewer #1: **Yes: **Paul F Zelkovic, MD

Reviewer #2: No

---

## [Author Response · Author response to Decision Letter 0]

8 Dec 2021

Reviewer #1: The submitted article presents useful data for clinical practice. The submitted statistical analysis is valid, showing diagnosis of GU tract anomalies to be rare. I do not routinely obtain upper tract imaging in patients presenting with orchitis, but there is debate about this, especially in younger boys. The submitted article adds to the evidence that upper tract imaging is not needed.

Dear Reviewer #1, thank you for your positive feedback. We hope too that the submitted article will contribute to taking forward discussion on OE management.

Reviewer #2: I found this to be a well written and clinically useful manuscript. In addition to the screening urinary tract US data, other findings in this patient population included a very low incidence of UTI (3.1%), inflammatory markers that were indeterminate and noncontributory, and 75% of patients had improved symptoms after 48 hours. In trying to understand this improvement, the role of treatment received from the ER in terms of antibiotic and pain management is unclear. Was there a consistent ER protocol for treatment? Were antibiotics given only when the urinalysis was positive? Was there a standard pain management regimen? The 3 documented UTIs occurred in infants less than 6 months of age. Although they had negative urinary tract ultrasounds, would you advise further evaluation with a VCUG in these patients after the first OE episode?

Minor corrections: The study dates given in the Abstract are 2011-2017 while in the Methods and Results the dates are 2012-2017. Also, in figure 1 the dates are 2011-2017.

Dear Reviewer #2, thank you for your positive feedback and questions.

Antibiotic and pain management: In our pediatric center, all children diagnosed with OE receive a prescription of paracetamol. Regarding antibiotic treatment, it is limited to urinary tract infections (positive urinanalysis). We have added this information in the manuscript. 

VCUG: In our center we follow the Swiss consensus recommendations on urinary tract infections in children. VCUG is only conducted when an anomaly is detected by a renal and urinary tract ultrasound or in case of recurrent UTI infections. We would not advise further evaluation with a VCUG after the first OE episode.

Minor corrections: the correct time frame is 2012-2017. We have corrected this information in the abstract and in figure 1.

---

## [Editor Report · Decision Letter 1]

31 Jan 2022

How useful is a complete urinary tract ultrasound in orchiepididymitis?

PONE-D-21-31965R1

Dear Dr. Aeschimann,

We’re pleased to inform you that your manuscript has been judged scientifically suitable for publication and will be formally accepted for publication once it meets all outstanding technical requirements.

Kind regards,

Israel Franco, M.D.

Academic Editor

PLOS ONE
---

## [Editor Report · Acceptance letter]

3 Feb 2022

PONE-D-21-31965R1 

How useful is a complete urinary tract ultrasound in orchiepididymitis? 

Dear Dr. Aeschimann:

I'm pleased to inform you that your manuscript has been deemed suitable for publication in PLOS ONE. Congratulations! Your manuscript is now with our production department. 

Kind regards, 

on behalf of

Dr. Israel Franco 

Academic Editor

PLOS ONE